# Long-term potentiation is independent of the C-tail of the GluA1 AMPA receptor subunit

Javier Díaz-Alonso[1†*], Wade Morishita[2], Salvatore Incontro[1], Jeffrey Simms[3], Julia Holtzman[3], Michael Gill[3], Lennart Mucke[3,4], Robert C Malenka[2], Roger A Nicoll[1*]

[1]Department of Cellular and Molecular Pharmacology, University of California, San Francisco, San Francisco, United States; [2]Nancy Pritzker Laboratory, Department of Psychiatry and Behavioral Sciences, Stanford University School of Medicine, Stanford, United States; [3]Gladstone Institute of Neurological Disease, San Francisco, United States; [4]Department of Neurology, University of California, San Francisco, San Francisco, United States

**Abstract** We tested the proposal that the C-terminal domain (CTD) of the AMPAR subunit GluA1 is required for LTP. We found that a knock-in mouse lacking the CTD of GluA1 expresses normal LTP and spatial memory, assayed by the Morris water maze. Our results support a model in which LTP generates synaptic slots, which capture passively diffusing AMPARs.

**\*For correspondence:**
jdiazalo@uci.edu (JD-A);
roger.nicoll@ucsf.edu (RAN)

**Present address:** [†]Department of Anatomy and Neurobiology, University of California, Irvine, Irvine, United States

**Competing interests:** The authors declare that no competing interests exist.

## Introduction

Long-term potentiation (LTP) requires the activity-dependent trafficking of AMPA receptors (AMPARs) to the synapse (*Collingridge et al., 2004*; *Malinow and Malenka, 2002*; *Nicoll, 2017*). Most AMPARs in CA1 pyramidal cells are heterotetramers of either GluA1/GluA2 subunits or GluA2/GluA3 subunits, although other complexes can also occur (*Zhao et al., 2019*). The prevailing, receptor centric, LTP model, posits that LTP-mediated covalent modification of the intracellular carboxy-terminal domain (CTD, also referred to as C-tail) of GluA1 results in the capture of these modified GluA1 containing receptors by preexisting 'slots' in the postsynaptic density (PSD) (*Hayashi et al., 2000*; *Huganir and Nicoll, 2013*; *Malinow and Malenka, 2002*; *Shi et al., 2001*), either by increasing the surface pool of AMPARs or the docking of AMPAR at the PSD. The nature of these slots is unclear, but it is thought to involve binding sites on postsynaptic scaffolding proteins, such as PSD-95. Two phosphorylation sites in the GluA1 CTD, S831 and S845, have received most of the attention. However, the occurrence of S831 and S845 phosphorylation in vivo is a matter of debate. A recent study found the relative abundance of phosphorylated GluA1 to be 'almost negligible' (*Hosokawa et al., 2015*), but see *Diering et al., 2016*. The replacement by alanine of either of these residues does not affect LTP (*Lee et al., 2010*), and only adult double phosphomutant mice have partially impaired LTP (*Lee et al., 2003*). In order to determine the minimal requirement for the GluA1 CTD during LTP, a previous study *Granger et al., 2013* used a conditional genetic knockout approach coupled with molecular replacement of AMPAR subunits. The Cre recombinase was transfected in CA1 pyramidal neurons in the hippocampus of *Gria1*, *Gria2* and *Gria3* floxed mice (*Gria1-3fl/fl*), in order to delete all endogenous AMPARs in *Lu et al., 2009*. We then expressed various modified GluA subunits upon this AMPAR null background. In the most relevant experiment in our study, we expressed a heteromeric receptor containing the GluA1 subunit lacking the CTD (GluA1ΔC) as well as GluA2, and observed normal basal trafficking and LTP at CA1 synapses in acute hippocampal slices. We therefore concluded that AMPAR lacking the GluA1 subunit CTD can traffic normally to

the synapse and enable normal LTP (*Granger et al., 2013*). These findings appear to be incompatible with the receptor centric model and the requirement of the GluA1 CTD for LTP.

A recent study has resurrected the receptor centric model of LTP (*Zhou et al., 2018*). The authors generated a knock-in (KI) mouse, in which they replaced the endogenous GluA1 with a chimeric GluA1 subunit that contains the CTD of GluA2 (GluA1A2CTD). They found that, while basal synaptic transmission was normal in this mouse, LTP was absent. Furthermore, a complementary chimeric AMPAR subunit, GluA2A1CTD, fully rescued LTP. Thus, the authors concluded that the CTD of GluA1 is 'necessary and sufficient' for NMDAR dependent LTP. What could explain this seeming contradiction? The present study addresses the discrepancy between the previous works (*Granger et al., 2013*; *Zhou et al., 2018*).

## Results

To address this discrepancy, we aimed to replicate the key experiments in Zhou et al. using overexpression and molecular replacement strategies (*Díaz-Alonso et al., 2017*; *Granger et al., 2013*). We previously showed that replacement of endogenous GluA2 subunits with GluA1/A2CTD resulted in functional AMPARs, which supported homeostatic synaptic scaling (*Ancona Esselmann et al., 2017*). Furthermore, overexpression of this construct in hippocampal slice cultures generated rectifying synaptic responses (*Figure 1—figure supplement 1A,B*), confirming that this construct forms functional homomeric receptors which traffic to the synapse constitutively. We next replaced all endogenous AMPARs with heteromeric GluA1/A2CTD-GluA2 receptors in hippocampal CA1 pyramidal neurons. To do so, we electroporated Cre recombinase in utero in *Gria1-3$^{f/f}$* mice (where all AMPAR subunits expressed in CA1 pyramidal neurons are floxed) together with GluA1/A2CTD and GluA2(R) (*Figure 1A*). Acute slices were prepared at P17-P25. Synaptic AMPARs were fully rescued (*Figure 1B*). Unlike replacement with GluA1/A2CTD alone, which results in strongly rectifying, homomeric AMPARs (*Figure 1—figure supplement 2B*), synaptic currents were non-rectifying in GluA1/A2CTD-GluA2(R) expressing neurons (*Figure 1—figure supplement 2A,C*), confirming that the expressed subunits form heteromeric receptors. These receptors exhibit normal LTP (*Figure 1C*). Trying to replicate the experiments reported by Zhou et al. more closely, we selectively replaced endogenous GluA1, which we deleted using CRISPR/Cas9 technology, with GluA1/A2CTD (*Figure 1D*). We initially tested the efficacy of the CRISPR/Cas9 guided GluA1 knockdown strategy in a heterologous system, 293 T cells. Co-transfection of a *Gria1* gRNA/Cas9 expressing vector in cells expressing GluA1 virtually eliminated the GluA1 protein (*Figure 1—figure supplement 3A*). We then tested the efficacy of the *Gria1* gRNA/Cas9 construct in hippocampal slices. Similar to the results obtained using the conditional KO of GluA1 using Cre-loxP (*Granger et al., 2013*; *Lu et al., 2009*), we observed a ~50% loss of AMPAR EPSCs when expressing the *Gria1* gRNA/Cas9 in rat slice cultures and mouse acute slices (therefore, data were pooled, *Figure 1—figure supplement 3B,C*). NMDAR EPSCs remained unchanged (*Figure 1—figure supplement 3D*). LTP was absent (*Figure 1—figure supplement 3E*), in agreement with previous results, likely due to the lack of a sufficient reserve pool of receptors (*Granger et al., 2013*; *Zamanillo et al., 1999*). The endogenous GluA1 was then replaced with GluA1/A2CTD*, where the sequence recognized by the *Gria1* gRNA was replaced by another which translates to the same protein sequence (*Figure 1D*, *Figure 1—figure supplement 3A*, Materials and methods). GluA1/A2CTD* expression rescued basal synaptic transmission (*Figure 1E*) and LTP (*Figure 1F*). NMDAR EPSCs were normal in transfected cells (*Figure 1—figure supplement 3F*).

The only remaining difference in the experimental approach between our study and that of Zhou et al. is that they used the endogenous promoter to express GluA1/A2CTD, while we used overexpression. Thus, to unequivocally assess the necessity of the GluA1 CTD for LTP, we generated a KI mouse where the endogenous GluA1 CTD is truncated (HA-ΔCTD GluA1, *Figure 2A*, *Figure 2—figure supplement 1*, Materials and methods). Any LTP present in this mouse must, therefore, be independent of the GluA1 CTD. A number of experiments confirmed that our KI mouse did, indeed, lack the GluA1 CTD. Western blots were performed using antibodies to the ATD of GluA1, the CTD of GluA1 and the HA tag in synaptosomal-enriched P2 fractions (*Figure 2B*). The HA tag, which we attached to the truncated C-terminus to identify the ΔCTD GluA1 subunit, is present in both the heterozygous and the homozygous KI mice, but, as expected, is absent from WT mice. The CTD directed antibody labeled the WT and heterozygous, but not the homozygous KI mouse. The ATD-

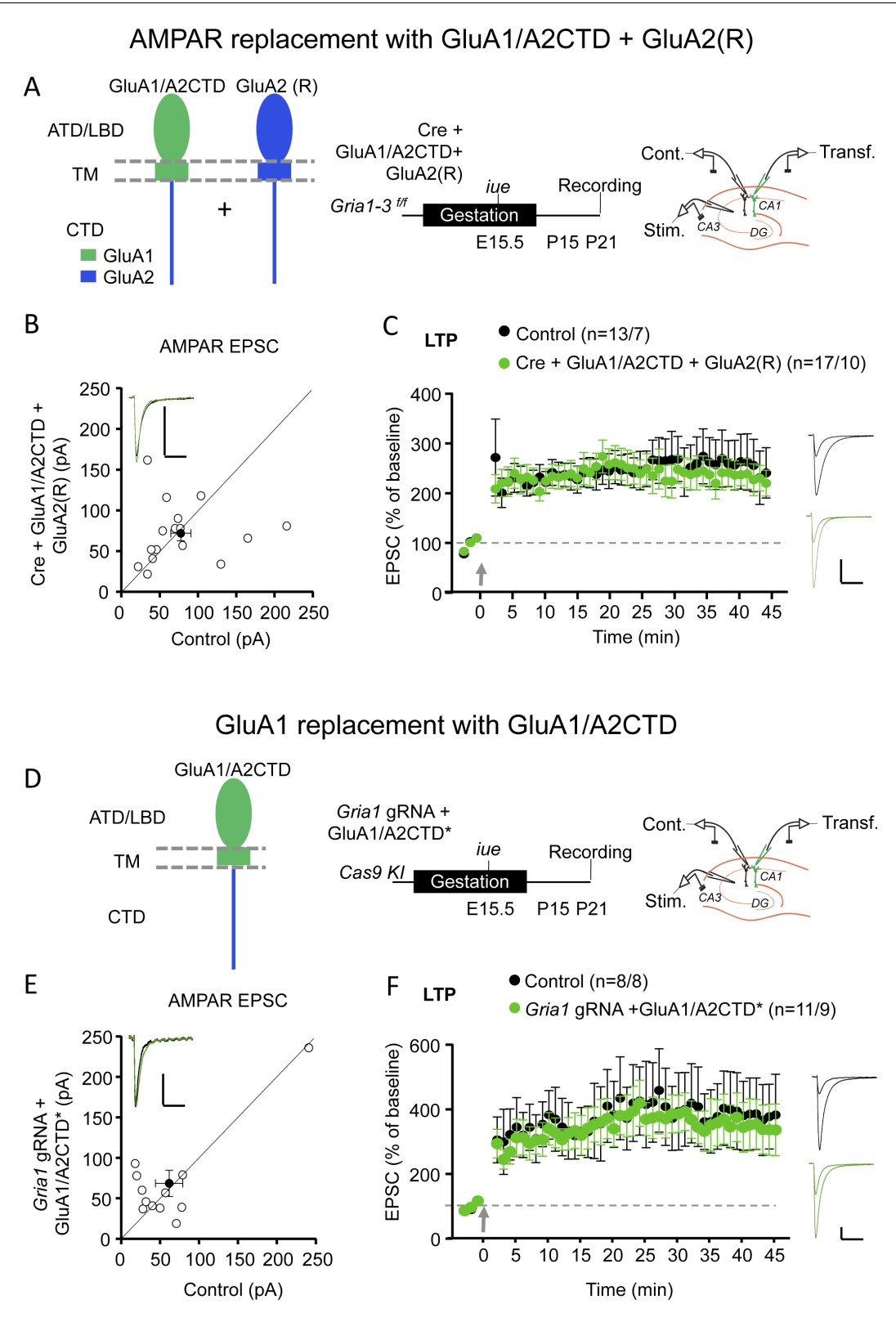

**Figure 1.** GluA1/A2CTD supports LTP. (**A**) Left panel, schematic illustration of the recombinant AMPAR subunits employed to replace endogenous AMPAR: GluA1/A2CTD and (edited) GluA2 (R) in hippocampal CA1 pyramidal neurons from *Gria1-3* $^{f/f}$ mice. Note that these two subunits form heteromeric, non-rectifying AMPAR (see *Figure 1—figure supplement 2*). ATD, amino-terminal domain; LBD, ligand-binding domain; TM, transmembrane domain; CTD, carboxy-terminal domain. Middle panel, summary and timeline of the experiment. Right panel, schematic illustration of

*Figure 1 continued on next page*

*Figure 1 continued*

the experimental setup with simultaneous whole-cell recordings from control and transfected CA1 pyramidal neurons. (B) Scatterplot measuring the baseline EPSC size at −70 mV in control (X axis) and Cre + GluA1/A2CTD + GluA2 (R) expressing (Y axis) neurons. Open circles represent individual pairs of control and transfected neurons, filled circle represents mean ± SEM. Inset shows sample traces from a control (black trace) and a transfected (green trace) cell. n = 16 pairs. p=0.804, two-tailed Wilcoxon signed-rank test. (C) Plot representing the mean ± SEM EPSC at −70 before and after LTP induction (arrow) normalized by the average baseline EPSC size (dashed gray line) in control (filled circles) and Cre + GluA1/A2CTD + GluA2 (R) expressing (green circles) CA1 pyramidal neurons. Sample traces before and 45' after LTP induction in control (black traces) and transfected (green traces) CA1 pyramidal neurons are shown to the right of the plot. n initial/final = 13/7 control, 17/10 transfected neurons. p=0.775 (min. 45), unpaired t-test. (D) Left panel, schematic illustration of the recombinant AMPAR subunit employed to replace endogenous GluA1: GluA1/A2CTD in hippocampal CA1 pyramidal neurons from Cas9 KI mice. Middle panel, summary and timeline of the experiment. Right panel, schematic illustration of the experimental setup with simultaneous whole-cell recordings from control and transfected CA1 pyramidal neurons. (E) Scatterplot measuring the baseline EPSC size at −70 mV in control (X axis) and *Gria1* gRNA + GluA1/A2CTD expressing (Y axis) neurons. Open circles represent individual pairs of control and transfected neurons, filled circle represents mean ± SEM. Inset shows sample traces from a control (black trace) and a transfected (green trace) cell. n = 12 pairs. p=0.557, two-tailed Wilcoxon signed-rank test. (F) Plot representing the mean ± SEM EPSC at −70 mV before and after LTP induction (arrow) normalized by the average baseline EPSC size (dashed gray line) in control (filled circles) and *Gria1* gRNA + GluA1/A2CTD expressing (green circles) CA1 pyramidal neurons. Sample traces before and 45' after LTP induction in control (black traces) and transfected (green traces) CA1 pyramidal neurons are shown to the right of the plot. Scale bars: 50 pA, 50 ms. n initial/final = 8/8 control, 11/9 transfected neurons. p=0.683 (min 45), unpaired t-test.

The online version of this article includes the following source data and figure supplement(s) for figure 1:

**Source data 1.** Contains source data for *Figure 1*.
**Figure supplement 1.** Chimeric GluA1/A2CTD traffics constitutively to the synapse.
**Figure supplement 1—source data 1.** Contains source data for *Figure 1—figure supplement 1*.
**Figure supplement 2.** GluA1/A2CTD and GluA2(R) form non-rectifying heteromeric AMPAR.
**Figure supplement 2—source data 1.** Contains source data for *Figure 1—figure supplement 2*.
**Figure supplement 3.** Validation of the CRISPR-mediated GluA1 deletion.
**Figure supplement 3—source data 1.** Contains source data for *Figure 1—figure supplement 3*.

directed antibody demonstrated the presence of GluA1 at normal levels in the KI mouse, where, as expected, the protein size is reduced due to the lack of the C-terminal 77 amino acids. Immunoblot against the GluA2 CTD and NR1 showed normal levels of these synaptic proteins in the KI (*Figure 2B*). Truncation of the GluA1 CTD was further confirmed with immunofluorescence using a GluA1 CTD antibody, which yielded strong staining in the WT hippocampal CA1 region, but no staining in the KI mouse (*Figure 2C*). AMPAR responses recorded from somatic outside out patches were unchanged in the KI mouse (*Figure 2D*). This is particularly important, because LTP expression is critically dependent on the level of extrasynaptic AMPARs (*Granger et al., 2013*). Furthermore, there was no change in the AMPAR/NMDAR ratio, consistent with a normal number of synaptic AMPARs (*Figure 2E*). Pairing-induced LTP (2 Hz/90 s. stimuli, while holding the postsynaptic neuron at 0 mV) in these KI mice was no different from WT controls (*Figure 2F*). To obtain an independent analysis of these mice, we collaborated with another group (R.C. Malenka and W. Morishita, Stanford University) who induced LTP with a different pairing protocol consisting of two stimulus bouts of 100 Hz/1 s. while holding the postsynaptic neuron at 0 mV. Again, no impairment in LTP was observed (*Figure 2G*).

In a final series of experiments, we tested hippocampal spatial learning and memory in these mice using the Morris water maze, a behavioral test that was shown to be impaired in GluA1A2CTD mice (*Zhou et al., 2018*). No statistically significant difference between WT and HA-ΔCTD GluA1 mice was found in either the distance travelled to find the hidden platform during training (*Figure 3A*), or in the ability to remember the position of the platform 24 hr after the last training session (*Figure 3B*, *Figure 3—figure supplement 1C*). HA-ΔCTD GluA1 mice showed a reduced swim speed across the training and test sessions (*Figure 3—figure supplement 1A,E*), which increased their latency to find the platform during training (*Figure 3—figure supplement 1B*), and resulted in a not significant trend toward increased latency to the first platform crossing in the probe trial (*Figure 3—figure supplement 1D*).

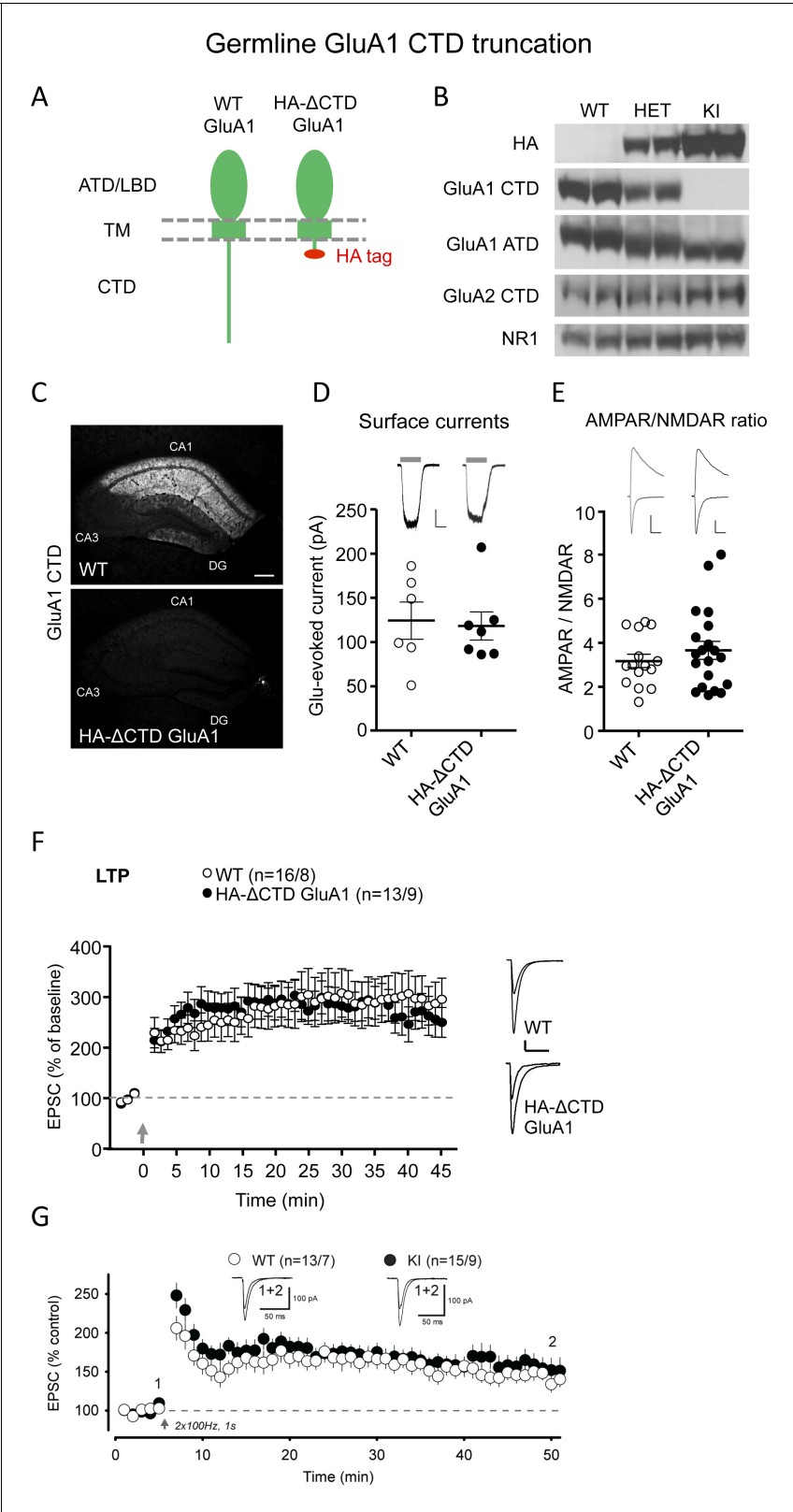

**Figure 2.** GluA1 CTD is not required for AMPAR trafficking and LTP. (**A**) Schematic illustration of WT GluA1 (left) and transgenic HA-ΔCTD GluA1 (right). The latter has the entire cytoplasmic tail truncated after the fourth amino acid after the last TM helix. ATD, amino-terminal domain; LBD, ligand-binding domain; TM, transmembrane domain; CTD, carboxy-terminal domain. (**B**) Western blots showing specific and allelic dose-dependent presence of haemmaglutinin (HA) tag only in heterozygous and homozygous HA-ΔCTD GluA1 mice brains, partial and total absence of signal from anti-GluA1

*Figure 2 continued on next page*

*Figure 2 continued*

CTD antibody in heterozygous and homozygous HA-ΔCTD GluA1 mice brains, respectively and decreased size of the GluA1 protein as a result of the truncation of the cytoplasmic tail in HA-ΔCTD GluA1 mice brains. GluA2 CTD and NR1 signals did not differ substantially among genotypes. Two biological replicates (mice) are shown. Three more mice per genotype were tested and several technical replicates were performed. (C) Assessment of the GluA1 CTD signal in the hippocampus of WT (top image) and HA-ΔCTD GluA1 (bottom image) mice by immunofluorescence. (D) Surface AMPAR-mediated currents elicited by fast glutamate (1 mM) application in WT (open circles) and HA-ΔCTD GluA1 (filled circles) hippocampal CA1 pyramidal neurons measured in somatic outside-out patches. Individual data values and mean ± SEM are indicated. Sample traces from WT (left) and KI (right) patches are shown to the top of the plot. Scale bars: 25 pA, 2 s. n = 6 WT and 7 HA-ΔCTD GluA1 KI patches. p=0.820, unpaired t-test. (E) AMPAR/NMDAR EPSC ratios measured at −70 mV and +40 mV (at 150 ms), respectively, in WT (open circles) and HA-ΔCTD GluA1 (filled circles) hippocampal CA1 pyramidal neurons. Individual data values and mean ± SEM are indicated. Sample traces from WT (left) and KI (right) neurons are shown to the top of the plot. Scale bars: 50 pA, 50 ms. n = 15 WT, 20 KI cells. p=0.377, unpaired t-test. (F) Plot representing the mean ± SEM EPSC at −70 mV before and after LTP induction (arrow) normalized by the average baseline EPSC size (dashed gray line) in WT (open circles) and HA-ΔCTD GluA1 KI (filled circles) CA1 pyramidal neurons. Sample traces before and 45' after LTP induction in WT (top) and KI (bottom) CA1 pyramidal neurons are shown to the right of the plot. Scale bars: 50 pA, 50 ms. n initial/final = 16/8 WT, 13/9 KI neurons. p=0.368 (min. 45). Unpaired t-test. (G) Plot representing the mean ± SEM EPSC at −70 mV before and after LTP induction (arrow) with an alternative protocol (2 bursts of 1 s duration at 100 Hz while holding the membrane potential at 0 mV) performed in an independent laboratory normalized by the average baseline EPSC size (dashed black line) in WT (n, cells/mice = 13/7, open circles) and HA-ΔCTD GluA1 KI (n, cells/mice = 15/9, filled circles) CA1 pyramidal neurons. Sample traces before LTP induction and at min. 50 in WT (left) and KI (right) CA1 pyramidal neurons are shown at the top of the plot at the indicated time points. p=0.606 (min 45 post pairing). Unpaired t-test.

The online version of this article includes the following source data and figure supplement(s) for figure 2:

**Source data 1.** Contains source data for *Figure 2*.
**Figure supplement 1.** Generation of the HA-ΔCTD GluA1 KI mouse line.

## Discussion

This study addressed whether the CTD of GluA1 is required for LTP and spatial memory, as recently reported (*Zhou et al., 2018*). We were unable to replicate these previous LTP results when we replaced endogenous GluA1 with GluA1/A2CTD using in utero electroporation. To test whether the high expression levels of the GluA1/A2CTD construct achieved by overexpression were masking LTP deficits, we generated a more conclusive KI mouse model. Instead of knocking in GluA1/A2CTD, as was done in the previous study, we truncated the CTD of the endogenous GluA1 (HA-ΔCTD GluA1) after the EFCY sequence following the last transmembrane helix of the polypeptide. Of note, this sequence is homologous in GluA1 and GluA2, so there is virtually no GluA1-specific CTD in this mouse. No defect in basal synaptic transmission, LTP, or spatial memory was found. What could account for the different results? By design, our LTP induction protocol is nearly-saturating, so that we can identify key, essential components of LTP. It is possible that a weaker induction protocol could reveal some subtle defects caused by the lack of the GluA1 CTD. However, Zhou et al. used a protocol similar to ours (cesium based internal solution with 100 Hz/1 s. tetanus repeated four times, see their Figure 7b). This LTP induction protocol would be at least as strong as ours, but they found no LTP in their GluA1A2CTD KI mouse.

Given that GluA1 KO mice show no spatial learning defects in the Morris water maze (*Zamanillo et al., 1999*), it is not surprising that mice lacking the GluA1 CTD did not show a spatial learning impairment in our study either. The severe deficits found in spatial learning and memory in GluA1/A2CTD KI mice are, therefore, puzzling. Of note, both GluA1 KO (*Zamanillo et al., 1999*) and HA-ΔCTD GluA1 (*Figure 3—figure supplement 1A,E*) mice show decreased swim speed compared to their WT controls, a possible confounding factor suggesting that the GluA1 CTD might be involved in this locomotor function. Future research will allow the dissection of the precise role played by the GluA1 CTD in locomotion and spatial memory, as well as other physiological and behavioral functions.

A large body of research has suggested that the GluA1 CTD modulates AMPAR trafficking and synaptic plasticity. We refer to these findings as the 'receptor centric' model of LTP, in which the LTP signaling pathway, presumed to involve CaMKII, targets the receptor, increasing the capture of modified receptors by preexisting slots in the PSD. Although we failed to find a fundamental requirement for the GluA1 CTD in AMPAR synaptic transmission, LTP and spatial memory, it has previously been shown that posttranslational modifications targeting this domain are involved in the modulation of these phenomena, particularly LTP. Multiple reasons might explain the apparent

conflict between our results and previous research. Perhaps, the well-established phosphorylation of GluA1 C-tail residues (particularly S831 and S845) (*Barria et al., 1997*; *Esteban et al., 2003*; *Hayashi et al., 2000*; *Lee et al., 2003*; *Mammen et al., 1997*; *Roche et al., 1996*) is crucial to relieve some, yet unidentified, negative modulatory effect exerted by other part(s) of the GluA1 C-tail, this negative modulation being absent in HA-ΔCTD GluA1 mice and in cells expressing GluA1/A2CTD. Our study was designed to assess the necessity of the GluA1 CTD in hippocampal LTP. Our data indicate that LTP does not require the GluA1 CTD and is, therefore, consistent with a model where LTP can occur independently of the subunit composition of AMPAR, in agreement with a previous study (*Granger et al., 2013*). More broadly, our results suggest an alternative model, which we refer to as the 'PSD centric model' for LTP, in which the LTP signal creates/unmasks new slots in the PSD that capture passively diffusing, unmodified AMPARs.

Based on recent findings from us and others (*Díaz-Alonso et al., 2017*; *Sheng et al., 2018*; *Watson et al., 2017*; *Watson et al., 2020*; *Zeng et al., 2019*), we propose that constitutive and activity-dependent AMPAR trafficking has two essential requirements. On one hand, the multivalent interaction between transmembrane AMPAR regulatory proteins (TARPs) and PSD scaffolding proteins (the intracellular slot). On the other hand, the presence of the GluA1 amino-terminal domain

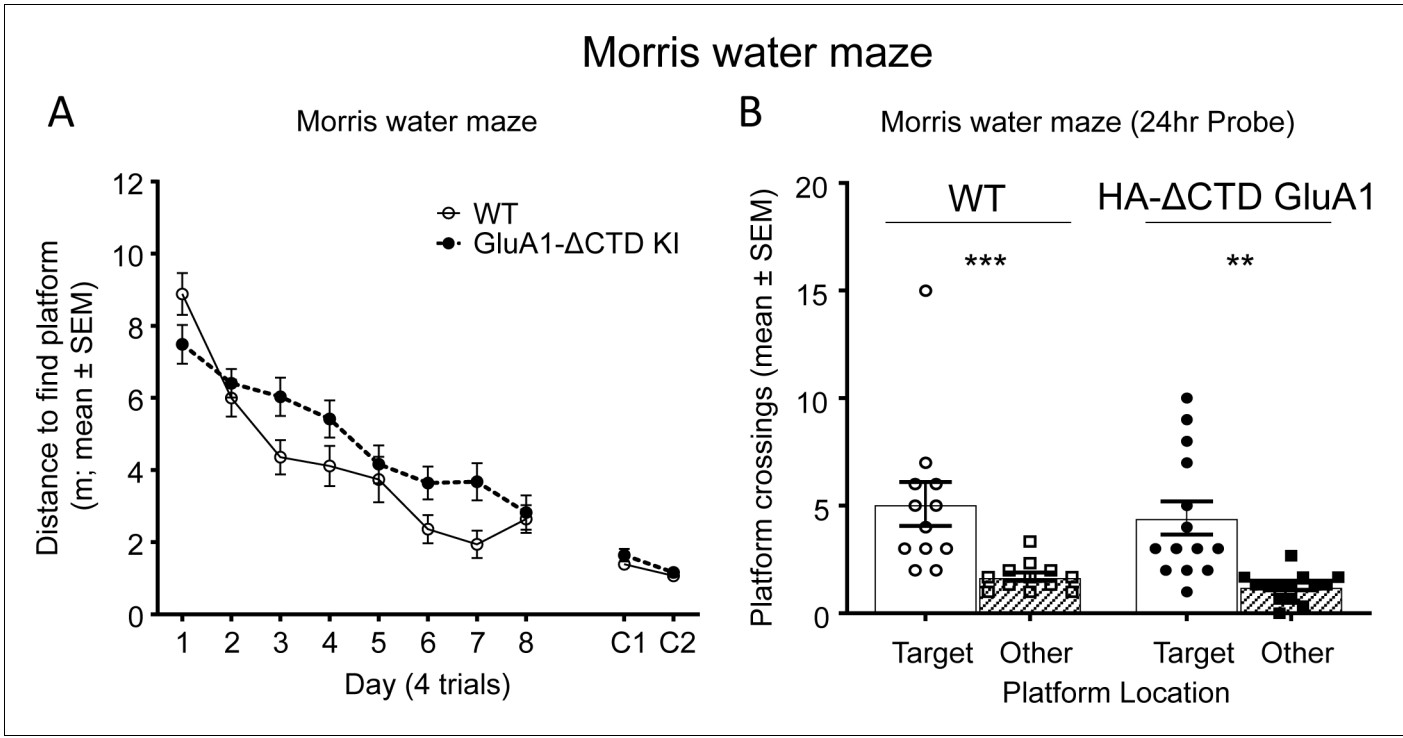

**Figure 3.** GluA1 CTD is not essential for spatial learning and memory. (**A**) Learning curves showing the distance covered to find a hidden platform in the Morris water maze per training day (average of 4 trials/day) in WT (open circles) and HA-ΔCTD GluA1 KI (filled circles) mice. Mixed effects analysis revealed that the distance necessary to find the platform decreased during training in both groups (day effect, p<0.0001). Although there was only a trend toward a genotype effect (p=0.0539), there was a significant interaction between day and genotype (p<0.05). Distance covered to find a cued platform across 2 days (C1 and C2) is shown in the right side of the plot and showed a significant effect of day (p<0.01) but not genotype (p=0.259), and there was no significant day x genotype interaction (p=0.511). n = 12 WT, 15 KI. (**B**) Probe trial results showing the number of crossings over the location under which the platform was hidden in the target quadrant during training (circles, empty bars) and over equivalent positions in non-target quadrants (squares, patterned bars) in a 60-s trial performed the day after the last training session. WT mice are represented by empty shapes and HA-ΔCTD GluA1 KI by filled shapes. n = 12 WT, 14 KI. Both genotypes showed a clear preference for the target location vs non-target locations (WT, p=0.0010, KI, p=0.0012,). WT and KI mice did not differ significantly in how many times they crossed the target location (p=0.582 by Mann-Whitney U test). **p<0.01; ***p<0.001 by Wilcoxon paired t-test. Individual mouse values and mean ± SEM are indicated.

The online version of this article includes the following source data and figure supplement(s) for figure 3:

**Source data 1.** Contains source data for *Figure 3*.

**Figure supplement 1.** Additional Morris water maze measurements.

**Figure supplement 1—source data 1.** Contains source data for *Figure 3—figure supplement 1*.

and its interaction with yet to be identified extracellular synaptic cleft moieties (the extracellular slot). This emerging model predicts that the activity-regulated availability of both intracellular and extracellular slots can modulate the abundance of functional AMPARs at the synapse.

# Materials and methods

## Key resources table

| Reagent type (species) or resource | Designation | Source or reference | Identifiers | Additional information |
|---|---|---|---|---|
| Gene (*Mus musculus*) | Gria1 | GenBank | #14799 | |
| Strain, strain background (*Mus musculus*, strain C57BL6) | HA-ΔCTD GluA1 | This paper | N/A | |
| Strain, strain background (*Mus musculus*, strain FVB) | Rosa26-Cas9 KI | The Jackson Laboratory | #026558; RRID:IMSR_JAX:026558 | |
| Strain, strain background (*Mus musculus*, strain C57BL6) | Gria1-3<sup>f/f</sup> | *Lu et al., 2009* | N/A | |
| Strain, strain background (*Rattus norvegicus*, strain CD Sprague Dawley IGS) | WT | Charles River | #001 | |
| Cell line (*Homo sapiens*) | 293T | ATCC | #CRL-3216; RRID:CVCL_0063 | |
| Recombinant DNA reagent (plasmid) | pCAGGS-GluA1/A2 CTD-IRES-GFP | This paper | N/A | Expression of chimeric GluA1/A2 CTD and GFP (under IRES). |
| Recombinant DNA reagent (plasmid) | pCAGGS-GluA1/A2 CTD-IRES-mCherry* | This paper | N/A | Gria1 CRISPR-resistant expression of chimeric GluA1/A2 CTD and mCherry (under IRES) |
| Recombinant DNA reagent (plasmid) | pCAGGS-IRES-mCherry | *Incontro et al., 2014* | N/A | Expression of mCherry (under IRES) |
| Recombinant DNA reagent (plasmid) | pFUGW-Cre:GFP | *Díaz-Alonso et al., 2017* | N/A | Expression of Cre:GFP fusion protein |
| Recombinant DNA reagent (plasmid) | px458- Gria1-CRISPR | This paper | N/A | Expression of Gria1 gRNA, Cas9 and GFP. Derived from px458 vector (Addgene #48138 RRID:Addgene_48138) |
| Recombinant DNA reagent (plasmid) | px458- Grin1-CRISPR | *Incontro et al., 2014* | N/A | Expression of Grin1 gRNA, Cas9 and GFP. Derived from px458 vector (Addgene #48138); RRID:Addgene_48138 |
| Recombinant DNA reagent | ssDNA encoding HA tag and stop codons flanked by 60 bp long homology arms for HDR | This paper | N/A | Obtained from IDT. Injected in fertilized zigotes for HA-ΔCTD GluA1 KI mouse generation (see Materials and methods for sequence) |

*Continued on next page*

*Continued*

| Reagent type (species) or resource | Designation | Source or reference | Identifiers | Additional information |
|---|---|---|---|---|
| Other (Recombinant RNA reagent) | Gria1 1 gRNA | This paper | N/A | Obtained from IDT. Injected in fertilized zigotes for HA-ΔCTD GluA1 KI mouse generation (see Materials and methods for sequence) |
| Other (Recombinant RNA reagent) | Gria1 2 gRNA | This paper | N/A | Obtained from IDT. Injected in fertilized zigotes for HA-ΔCTD GluA1 KI mouse generation (see Materials and methods for sequence) |
| Antibody | Rabbit polyclonal anti-GluA1 C-tail | Synaptic Systems | #182–003; RRID:AB_2113441 | IF (1:500) WB (1:1000) |
| Antibody | Rabbit polyclonal anti-GluA2 C-tail | Synaptic Systems | #182–103; RRID:AB_2113732 | WB (1:1000) |
| Antibody | Mouse monoclonal anti-GluA1 ATD | Millipore | #MAB 2263; RRID:AB_11212678 | WB (1:1000) |
| Antibody | Mouse, monoclonal anti-NR1 | Millipore | #05–432; RRID:AB_390129 | WB (1:1000) |
| Antibody | Rabbit polyclonal anti-HA | Thermo Fisher Scientific | #71–5500; RRID:AB_2533988 | WB (1:1000) |
| Antibody | Rabbit polyclonal anti-alpha tubulin | Cell Signaling | #2144; RRID:AB_2210548 | WB (1:1000) |
| Antibody | HRP conjugated anti-mouse secondary antibody | GE Healthcare | #NA931; RRID:AB_772210 | WB (1:5000) |
| Antibody | HRP conjugated anti-rabbit secondary antibody | GE Healthcare | #NA934; RRID:AB_772206 | WB (1:5000) |
| Antibody | Alexa-488 conjugated anti-rabbit secondary antibody | Thermo Fisher Scientific | #A11034; RRID:AB_2576217 | IF (1:500) |
| Chemical compound, drug | D(-)—2-amino-5-phosphonovaleric acid (AP5) | Hello Bio | #HB0225 | 0.1 mM |
| Chemical compound, drug | Picrotoxin | TCI | #C0375 | 0.1 mM |
| Chemical compound, drug | Bicuculline | Sigma-Aldrich | #14340 | 0.02 mM |
| Chemical compound, drug | 2-Chloroadenosine | Sigma-Aldrich | #C5134 | 2 mM |
| Commercial assay, kit | Helios Gene Gun Kit | Bio-Rad | #1652411 | Used for biolistic transfection of hioppocampal slice cultures |
| Commercial assay, kit | In fusion HD cloning kit | Takara Bio | #639647 | Used for clonning of GluA1/A2 CTD in pCAGGS vectors |

*Continued on next page*

*Continued*

| Reagent type (species) or resource | Designation | Source or reference | Identifiers | Additional information |
|---|---|---|---|---|
| Commercial assay, kit | NheI | New England Biolabs | #R0131 | Restriction enzyme. Used for cloning of GluA1/A2 CTD in pCAGGS vectors |
| Commercial assay, kit | XhoI | New England Biolabs | #R0146 | Restriction enzyme. Used for cloning of GluA1/A2 CTD in pCAGGS vectors |
| Commercial assay, kit | BbsI | New England Biolabs | #R3539 | Restriction enzyme. Used for cloning of gRNA in px458 vectors |
| Commercial assay, kit | T4 DNA ligase | New England Biolabs | #M0202L | Ligase. Used for cloning of gRNA in px458 vectors |
| Commercial assay, kit | MycoAlert PLUS Mycoplasma Detection Kit | Lonza | #LT07-701 | Mycoplasma contamination assay |
| Commercial assay, kit | Lipofectamine 2000 | Thermo Fisher Scientific | #11668027 | Transfection reagent for 293 T cells |
| Software, algorithm | Prism | Graph Pad | https://www.graphpad.com/scientific-software/ prism/; RRID:SCR_002798 | |
| Software, algorithm | Igor Pro | Wavemetrics | https://www.wavemetrics.com/products/igorpro; RRID:SCR_000325 | |
| Software, algorithm | ImageJ | NIH | https://imagej.nih.gov/ij/; RRID:SCR_003070 | |

## Animals

All animal procedures were approved by the Institutional Animal Care and Use Committee at the University of California, San Francisco (protocol numbers AN170318 and AN183289) and Stanford (protocol number 10322). All animals were maintained in 12 hr light/dark schedule and with access to food and water, ad libitum.

### Generation of HA-ΔCTD GluA1 mice

Super-ovulated female C57BL/6 mice (4 weeks old) were mated to C57BL/6 stud males. Fertilized zygotes were collected from oviducts and injected with Cas9 protein (30 ng/µl), crRNA (20 ng/µl) tracrRNA (20 ng/µl), and ssDNA (10 ng/µl) into the pronucleus of fertilized zygotes. Two different crRNA sequences were designed using the gRNA design tool and purchased from Integrated DNA Technologies Inc:

1. **CAUCCGCUUCGACUCGCUAC**.
2. **UUUGUAGCAGAACUCGAUUA**.

Half of the embryos were injected with each one of the gRNAs and both generated transgenic mice. Therefore, we selected as founder a mouse modified with CAUCCGCUUCGACUCGCUAC, which had slightly better selectivity rating in the IDT gRNA design tool.

A ssDNA encoding the influenza haemagglutinin (HA) tag sequence followed by four Stop codons flanked by 60 nt long 5' and a 3' homology arms was designed to provide a template for homology-directed repair (HDR) in CRISPR/Cas9-edited zygotes and purchased from Integrated DNA Technologies Inc with the following sequence: TACATCCTGATTGGAGGGCTGGGATTGGCCATGCTGG

TTGCCTTAATCGAGTTCTGCTAC**TACCCATACGATGTTCCAGATTACGCT***TAATAGTGATA*AAAA
TCCCGTAGCGAGTCGAAGCGGATGAAGGTGGCATCGTCTTCCCGGATCTTTTCCCTA (HA
sequence is bolded and stop codons are in italics).

Injected zygotes were implanted into oviducts of pseudopregnant CD1 female mice. Successful transgenesis was assessed in the F1 mice by sequencing and genotyping. Several heterozygous F1 mice were identified where insertion of the HA-Stop sequence had happened in the appropriate site. One was chosen as the founder of the colony and backcrossed at least three generations before used for experiments. Genotyping was performed by TransnetYX INC. USA, after assessing that their assay provided results 100% identical to sequencing. For electrophysiology experiments, male and female mice 17–25 days of age (Nicoll lab) and 30–45 days of age (Malenka lab) were used. For behavior experiments, 3–4 months of age male littermates and cage mates generated by heterozygous breedings and homozygous WT and HA-ΔCTD GluA1 KI breedings, respectively, were used. For western blot and immunofluorescence, 90 day-old males and females were used.

*Gria1-3* $^{f/f}$ mice used in AMPAR replacement experiments were generated and genotyped as described previously (*Lu et al., 2009*).

Rosa26-Cas9 KI mice used in GluA1 replacement experiments were purchased from The Jackson Laboratory and maintained as previously described (*Platt et al., 2014*).

P6-8 rat pups were employed to generate the organotypic hippocampal slice cultures employed in GluA1/A2CTD overexpression experiments as described previously (*Stoppini et al., 1991*).

## Cells

293 T cells were purchased from ATCC and maintained in DMEM (Gibco) with 10% FBS (GenClone). Cells were passaged a maximum of four times after thawing the original vial from ATCC. Mycoplasma infection was assessed with MycoAlert PLUS Mycoplasma Detection Kit (Lonza).

## Constructs

The gRNA for acute deletion of *Gria1* was designed as previously described (*Incontro et al., 2014*), using the MIT online design tool CRISPR/Cas9 (http://crispr.mit.edu) and subcloned into the human codon-optimized Cas9 and chimeric gRNA expression plasmid px458 (Addgene, *Ran et al., 2013*) using T4 DNA ligase. The gRNA sequence selected (forward, 5' to 3': GACCATAACCTTGG TCCGGG; reverse, 5' to 3': CCCGGACCAAGGTTATGGTC) is specific for *Gria1* and shared by rat and mouse. px458 *Grin1 g*RNA (*Incontro et al., 2014*) was used as a control.

GluA1/A2CTD was subcloned into a pCAGGS-IRES-GFP and pCAGGS-IRES-mCherry vectors from a pFUGW used in previous work (*Ancona Esselmann et al., 2017*) using the In-Fusion HD Cloning System (Takara Bio, USA, Inc). CRISPR-resistant pCAGGS-GluA1/A2CTD*-IRES-mCherry was generated by replacing by PCR the rat/mouse *Gria1* gRNA targeting sequence ACCATAACCTTGG TCCGG with the ACAATTACAATAGTGCGC sequence, which translates to the same amino acid sequence, expresses at similar levels and is not recognized by the *Gria1* gRNA (*Figure 1—figure supplement 3A*).

## Neuronal transfection

Biolistic transfection of organotypic slice cultures was performed as previously described (*Schnell et al., 2002*). In brief, 1-μm-diameter gold particles (Bio-Rad) were coated with 50 μg of pCAGGS-GluA1/A2CTD-IRES-GFP for overexpression experiments or px458 *Gria1* together with pCAGGS-IRES-mCherry to facilitate identification of transfected cells (GFP signal from the px458 construct is dim in our hands) for GluA1 knock-down experiments in 0.5 mM spermidine. DNA was then precipitated with 0.1 mM $CaCl_2$, and then gold particles washed three times in 100% ethanol. The gold particles were loaded onto PVC tubing (BioRad) and dried using ultra-pure N2 gas. DNA-coated gold particles were shot with a Helios GeneGun (Bio-Rad). Expression of recombinant GluA1/A2CTD was confirmed by GFP fluorescence.

In utero electroporation and in vivo AMPAR replacement. In utero electroporation was performed as previously described (*Díaz-Alonso et al., 2017*; *Navarro-Quiroga et al., 2007*). Briefly, E15.5 pregnant *Gria1-3* $^{f/f}$ or Cas9 KI female mice were anesthetized with 2% isoflurane in $0_2$. Buprenorphine (Reckitt Benckiser Healthcare) and meloxicam (Boehringer Ingelheim) were administered subcutaneously. 1.5 μl of plasmid DNA with Fast Green (Sigma Aldrich) were injected into the lateral

ventricles. In AMPAR replacement experiments, pFUGW-Cre:GFP was diluted to approximately 0.5 µg/µl and mixed with 2 µg/µl of the replacement pCAGGS-GluA1/A2CTD-IRES-GFP and pCAGGS-GluA2(R)-IRES-GFP plasmids. In GluA1 knock-down experiments, px458 *Gria1* gRNA was diluted to approximately 0.5 µg/µl and mixed with 2 µg/µl pCAGGS-IRES-mCherry (pCAGGS-GluA1/A2CTD*-IRES-mCherry in replacement experiments). Then, 5 × 40 V pulses of 50 ms. were delivered at 1 Hz, using platinum tweezertrodes in a square-wave pulse generator (BTX Harvard Apparatus). The positive electrode was placed in the lower right hemisphere and the negative electrode placed in the upper left hemisphere to direct transfection preferentially to the CA1 region of the hippocampus (*Navarro-Quiroga et al., 2007*). Following electroporation, embryos were returned to the abdominal cavity and abdominal muscle and skin were sutured. Complete recovery was ensured before returning females to their cage.

## Electrophysiology

Voltage-clamp recordings from CA1 pyramidal neurons were obtained using mouse acute hippocampal slices or rat organotypic slice cultures. 300 µm transverse acute slices were prepared with a Microslicer DTK-Zero1 (Ted Pella) in ice-cold high sucrose cutting solution containing (in mM): 2.5 KCl, 7 MgSO$_4$, 1.25 NaH2PO$_4$, 25 NaHCO$_3$, 7 glucose, 210 sucrose, 1.3 ascorbic acid. Slices were then incubated during 30 min at 34°C in artificial cerebrospinal fluid (aCSF) containing (in mM): 119 NaCl, 2.5 KCl, 1 NaH2PO$_4$, 26.2 NaHCO$_3$ and 11 glucose and allowed to recover at room temperature for 45 min. The aCSF was bubbled with carbogen (95% O$_2$/5% CO$_2$). For acute slices, 2.5 mM CaCl$_2$ and 1.3 mM MgSO$_4$ were added to the aCSF, and for organotypic slice cultures 4 mM CaCl$_2$ and 4 mM MgSO$_4$. During recording, slices were transferred to a perfusion stage on an Olympus BX51WI upright microscope and perfused at approx. 2.5 ml / min with aCSF containing 0.1 mM picrotoxin and 0.02 mM bicuculline to block GABA$_A$ transmission. 2 mM 2-Chloroadenosine was added to aCSF in experiments with slice cultures to manage epileptiform activity. The internal whole-cell recording solution contained (in mM) 135 CsMeSO$_4$, 8 NaCl, 10 Hepes, 0.3 EGTA, 5 QX-314, 4 Mg-ATP, and 0.3 Na-GTP and 0.1 spermine. Osmolarity was adjusted to 292 mOsm, and pH at 7.4. Synaptic responses were evoked with a bipolar tungsten stimulation electrode (Microprobes) placed in the striatum radiatum, at 0.2 Hz (basal transmission) or 0.1 Hz (LTP experiments). For the Stanford group, acute slice preparation and maintenance were similar with minor differences to the following. Transverse hippocampal slices (225 µm thick) were prepared with a vibratome (Leica VT1000s) in high sucrose cutting solution, which comprised (in mM): 2.5 KCl, 8 MgSO$_4$, 1.25 NaH2PO$_4$, 26.2 NaHCO$_3$, 20 glucose, 225 sucrose, 0.5 CaCl$_2$. Whole-cell recordings were performed in a perfusion chamber mounted on a fixed stage of an Olympus BX 50 WI microscope. Slices were perfused at approx. 1 ml/min with warm (30°C) oxygenated (95% O$_2$/5% CO$_2$) aCSF containing 50 µM picrotoxin. The internal whole-cell recording solution contained (in mM) 135 CsMeSO4, 8 NaCl, 10 HEPES, 0.25 EGTA, 2 MgCl$_2$, 5 phosphocreatine, 4 Mg-ATP and 0.3 Na-GTP (298–301 mOsM, pH 7.4). Membrane holding current, input resistance, and pipette series resistance were monitored throughout recordings. Data were gathered through a MultiClamp 700B amplifier (Axon Instruments), filtered at 2 kHz, and digitized at 10 kHz.

## Whole-cell synaptic recordings and LTP

AMPAR-mediated responses were isolated by voltage-clamping the cell at −70 mV, whereas NMDAR-mediated responses were recorded at +40 mV and measured at 150 ms after stimulation to avoid contribution of AMPAR. To calculate synaptic AMPAR rectification, 0.1 mM D(-)−2-amino-5-phosphonovaleric acid (AP5) was washed-in to block NMDARs. Rectification of synaptic responses was calculated as follows: RI = 7(I40 − I0)/4(I0 − I-70) where Ix represent EPSC amplitude at x mV.

Transfected cells were identified by their GFP or mCherry fluorescence. In simultaneous whole cell experiments, control, untransfected cells adjacent to the transfected cells were patched and recorded simultaneously.

LTP was induced, after recording a stable 3–5 min baseline, but not more than 6 min after breaking into the cell, by stimulating Schaffer collateral axons using two alternative protocols. In the Nicoll lab stimulation is at 2 Hz for 90 s, while in the Malenka lab it is 2 × 1 s at 100 Hz, while clamping the cell at 0 mV in both cases.

## Behavior

The Morris water maze test was performed as described in *Orr et al., 2018*. The water maze consisted of a 122 cm-diameter pool filled with water (21 ± 1°C) made opaque with nontoxic white tempera paint. Distinct extra-maze cues surrounded the pool. Before hidden platform training, mice underwent one session of four pre-training trials in which they swam in a rectangular channel (15 cm ×122 cm) and mounted a square platform (14 × 14 cm) hidden 1.5 cm below the water surface in the middle of the channel. Mice that did not mount the platform were guided gently to it by the experimenter and were allowed to sit on it for 10 s before being removed by the experimenter.

Three days after pre-training, mice were trained in the circular water maze. For hidden platform training, the platform was submerged 1.5 cm below the surface. The platform location remained the same throughout training, but the drop location varied randomly among the four daily trials. Mice received two sessions per day (3 hr intersession interval between sessions) for 8 consecutive days. Each session consisted of two trials with a 15-min intertrial interval. The maximum time allowed per trial was 60 s. If a mouse did not find or mount the platform, it was guided to the platform by the experimenter. All mice were allowed to sit on the platform for 10 s after each training trial.

For the probe trial, the platform was removed and each mouse was allowed to swim for 60 s. The drop location for the probe trial was 180° from the platform location used during hidden platform training. After 60 s, mice were guided to the where the platform had been located during hidden training before removal from the pool. Mice were probed 1 day after the completion of hidden platform training.

After probe testing, cued (visible) platform training was performed using new platform locations and a clearly visible cue (a 15 cm striped pole on top of the platform). Mice received three sessions of two cued trials per session across two days (15-min interval between trials and 3-hr interval between sessions). Each cued platform session was to a different location in the pool. All behaviors wer recorded and analyzed with an Ethovision XT video tracking system (Noldus). Escape latencies, distance traveled, swim speeds, platform crossings and proximity to the platform were recorded automatically for subsequent analysis. One mouse was excluded from the probe trial due to extreme floating behavior and two mice were excluded from both training and the probe trial due to procedural learning deficits. Exclusions were done blind to the genotype.

## Immunoblotting

48 hr post-transfection with Lipofectamine 2000 (Invitrogen), 293 T cells were washed in PBS, pelleted and re-suspended directly in SDS-containing sample buffer. WT and HA-and GluA1 mice forebrain tissue was processed as previously described (*Bemben et al., 2014*). Tissue was collected in ice-cold PBS and homogenized in TEVP buffer containing 20 mM Tris-HCl (pH 7.5), 0.3 M sucrose, 5 mM EDTA and protease and phosphatase inhibitors (Roche). After centrifugation at 1000 g for 10 min, the supernatant was centrifuged at 10,000 g for 20 min to obtain the P2 fraction. The P2 fraction was then re-suspended in SDS-containing sample buffer. All samples were run in a PAGE-SDS electrophoresis. PVDF membranes were blocked with 5% blotting grade nonfat milk (Bio-Rad) in tris buffered saline buffer with 0.1% tween 20 (Acros). The following primary antibodies were used (1/1000) in western blot experiments: GluA1 CTD (rabbit Synaptic Systems, #182–003), GluA1 ATD (mouse, Millipore, #MAB 2263), HA (rabbit, Invitrogen, #71–5500), NR1 (mouse, Millipore, #05–432), GluA2 CTD (rabbit, Synaptic Systems; #182–103), α-Tubulin (rabbit, Cell Signaling; #2144). HRP-conjugated secondary antibodies raised against the appropriate species were used. Images were processed using ImageJ.

## Immunofluorescence

PFA fixed, 30-μm-thick coronal brain slices were obtained and processed for immunofluorescence analysis. Immunofluorescence was performed, after blockade with 5% goat serum, by overnight incubation at 4°C with a GluA1 CTD primary antibody (rabbit, Synaptic Systems, #182–003) followed by incubation with an Alexa 488 anti-rabbit secondary antibody (Invitrogen). Images were obtained using a Leica DMRB fluorescence microscope and processed with ImageJ.

## Sampling and statistics

Summarized data were presented in figures as mean ± SEM with n values representing, in all cases, the number of biological replicates (number of cells, pairs or mice in each data set, as indicated in figure legends). Sample size for all experiments was estimated according to the standards in the field (*Díaz-Alonso et al., 2017*; *Granger et al., 2013*; *Incontro et al., 2014*; *Orr et al., 2018*). Genotype blinding (masking) was used for behavior experiments. Electrophysiology experiments were performed without masking.

Data analysis was carried out in Igor Pro (Wavemetrics) Excel (Microsoft), and GraphPad Prism (GraphPad Software). Unpaired t-test or Mann-Whitney U test were used to assess statistical significance in experiments involving unpaired data. Two-tailed Wilcoxon signed-rank test for experiments using paired data. For Morris water maze experiments, mixed effects analyses were employed to assess the effect of genotype and training in hidden platform and cued platform location performance and swim speed, while number of platform crossings and % time in quadrant in the 24 hr probe were analyzed using paired t-test and Wilcoxon signed-rank test. For measures directly comparing probe performance between genotypes (latency to first platform crossing and swim speed), Welch's t-test and Mann-Whitney U test were used. LTP data in molecular replacement experiments was obtained from pairs of control and experimental neurons; however, some cells were lost during the experiment, as indicated in the LTP plot legends and figure legends. Consequently, the resulting datasets are a mix of interleaved and paired data, thus, comparisons were made using unpaired statistics. Statistical significance of LTP in HA-ΔCTD GluA1 *vs* WT mice experiments was also analyzed with unpaired statistics. All statistical significances were set as $*p<0.05$, $**p<0.01$, and $***p<0.001$.

## Acknowledgements

Samantha Ancona-Esselman (Nicoll lab) for the pFUGW-GluA1/A2CTD construct. Junli Zhang (Gladstone institutes) for technical assistance in the generation of the HA-ΔCTD GluA1 KI mouse line. Eric Dang (Nicoll lab) for technical assistance. The Gladstone Institutes Behavioral Core for behavioral testing of mice. Dr. Argentina Lario-Lago and members of the Nicoll lab for valuable discussion. P Seeburg and R Sprengel for the individual *Gria1–3*fl/fl mice. David Julius (UCSF) for access to the cryostat and fluorescence microscope. Research was supported by R01MH070957 and R01MH117139 (to RAN), P50MH086403 (to RCM) and K99MH118425 (to JD-A).

## Additional information

### Funding

| Funder | Grant reference number | Author |
|---|---|---|
| National Institute of Mental Health | K99MH118425 | Javier Díaz-Alonso |
| National Institute of Mental Health | R01MH070957 | Roger A Nicoll |
| National Institute of Mental Health | R01MH117139 | Roger A Nicoll |
| National Institute of Mental Health | P50MH086403 | Robert C Malenka |

The funders had no role in study design, data collection and interpretation, or the decision to submit the work for publication.

### Author contributions

Javier Díaz-Alonso, Conceptualization, Resources, Data curation, Formal analysis, Funding acquisition, Validation, Investigation, Visualization, Writing - original draft, Project administration; Wade Morishita, Data curation, Formal analysis, Validation, Investigation, Writing - review and editing; Salvatore Incontro, Validation, Investigation; Jeffrey Simms, Data curation, Formal analysis, Investigation, Writing - review and editing; Julia Holtzman, Formal analysis, Investigation; Michael Gill, Data curation, Formal analysis; Lennart Mucke, Data curation, Formal analysis, Supervision, Writing -

review and editing; Robert C Malenka, Funding acquisition, Project administration, Writing - review and editing; Roger A Nicoll, Conceptualization, Formal analysis, Supervision, Funding acquisition, Writing - original draft, Project administration

### Author ORCIDs
Javier Díaz-Alonso ⓘ https://orcid.org/0000-0002-4980-7441
Roger A Nicoll ⓘ https://orcid.org/0000-0002-6977-4632

### Ethics

Animal experimentation: The authors declare that this study has been performed strictly following all relevant laboratory animal use regulations according to approved institutional animal care and use committee (IACUC) protocols of the University of California, San Francisco (AN170318 and AN183289), and Stanford University (10322).

### Decision letter and Author response

Decision letter https://doi.org/10.7554/eLife.58042.sa1
Author response https://doi.org/10.7554/eLife.58042.sa2

## Additional files

### Supplementary files
• Transparent reporting form

### Data availability
All data generated or analysed during this study are included in the manuscript and supporting files.

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
