## [Decision Letter]

**Acceptance summary:**

Long-term potentiation (LTP) is the strengthening of synapses between neurons that is a cellular mechanism of memory formation. Classical LTP is mediated by trafficking of glutamate receptors into the synapse, but there has been debate about how this occurs. The authors use multiple genetic approaches to show that the cytoplasmic tail of a glutamate receptor subunit, the GluA1 AMPAR subunit, which has been implicated in the past, is not essential for LTP. These results provide insight into the fundamental requirements for LTP and helps differentiate between models of receptor trafficking.

**Decision letter after peer review:**

Thank you for submitting your article "Long-term potentiation and spatial memory are independent of the C-tail of the GluA1 AMPA receptor subunit" for consideration by *eLife*. Your article has been reviewed by three peer reviewers, including Linda Overstreet-Wadiche as the Reviewing Editor and Reviewer #1, and the evaluation has been overseen by Richard Aldrich as the Senior Editor.

The reviewers have discussed the reviews with one another and the Reviewing Editor has drafted this decision to help you prepare a revised submission.

As the editors have judged that your manuscript is of interest, but as described below that additional experiments or major revisions are required before it is published, we would like to draw your attention to changes in our revision policy that we have made in response to COVID-19 (https://elifesciences.org/articles/57162). First, because many researchers have temporarily lost access to the labs, we will give authors as much time as they need to submit revised manuscripts. We are also offering, if you choose, to post the manuscript to bioRxiv (if it is not already there) along with this decision letter and a formal designation that the manuscript is "in revision at *eLife*". Please let us know if you would like to pursue this option. (If your work is more suitable for medRxiv, you will need to post the preprint yourself, as the mechanisms for us to do so are still in development.)

Summary:

This is an important study investigating a central question in synaptic neuroscience about mechanisms of AMPAR trafficking in LTP. Using three genetic approaches the authors convincingly show that the cytoplasmic tail of the GluA1 AMPAR subunit, which has been implicated in the past, is not essential. The reviewers agreed that the electrophysiological experiments are sophisticated, carefully executed, and informative. However, the behavioral tests are not as thorough as needed to support the claim that spatial memory is independent of the C-terminal domain. The authors should remove these data and claims, or provide additional experiments to make them more convincing.

Essential revisions:

1) Although the manuscript is clearly written, the authors need to provide more background to make this work accessible to a wider audience. In its current format one needs to be well-versed in the field and familiar with the Zhou et al. study to appreciate the new data. Please add explanation of “receptor slots”, more background on receptor-centric vs PSD-centric model (perhaps cite their recent review), the Malinow model, the role of receptor availability and the passive role for AMPARs in synaptic plasticity.

2) While the authors state that a widely held viewpoint is that covalent modifications of the GluA1 C-terminus results in GluA1 capture and anchoring at the postsynaptic site, this isn't necessarily the most prevalent model. Another view (not contradictory to the authors viewpoint that AMPARs in a reserve pool is critical for LTP) is that C-terminal modifications augment trafficking of GluA1-containing AMPAR to the postsynaptic site where via other mechanisms AMPAR become trapped to augment their postsynaptic functional availability. Framing the debate strictly in terms of receptor versus PSD centric models doesn't account for all views.

3) The text should be modified to clarify that the authors have not actually replicated the key experiments of Zhou et al., since they did not analyze the same mutations (replacing the GluA1 C-term with that of GLuA2 versus truncation) nor most of the same protocols (LTP of fEPSPs with inhibition intact, AMPAR conductance). Framing the current work primarily as addressing discrepancies with prior studies is not really satisfying when the discrepancies have not been explained.

4) There is no question that simultaneously mutating S831 plus S845 to alanine residues in the C-terminus of GluA1 in KI mice affects LTP induced by a so-called theta burst protocol as well as a pairing protocol, the latter similar to the protocols used in the current work (Lee et al., 2003, as cited). Apparently, when the GluA1 C-terminus is largely intact except for these two phosphorylation sites, then phosphorylation of one of the two sites is important for LTP (and learning as also tested in Lee et al., 2003). A possible explanation is that truncating the GluA1 C-terminus creates a situation in which such phosphorylation might be less important. For instance, perhaps non-phosphorylated GluA1 is held back from surface insertion or lateral diffusion to the postsynaptic site and phosphorylation of either S831 or S845 releases GluA1 to traffic to the postsynaptic site for anchoring there by means independent of S831 or S845 phosphorylation. When the C-terminus is removed as in the current manuscript than this inhibition is also removed. In order to put their results in context of prior work, the authors should discuss these possibilities and that the results depend on the exact mutations that are analyzed, and further the A2C-tail could also be an important factor.

5) There are several issues with the behavior data that raise concerns with the strong conclusion about spatial memory. The authors should remove these data and claims, or provide additional experiments to make them more convincing.

First, it would be more convincing to assess additional tests of spatial memory beyond MWM (such as novel object location and Y maze novel arm entry), especially in light of the swim speed confound.

Second, there are clear differences in performance during the training sessions in the MWM that warrant careful consideration. While the KI mice demonstrate gross learning day-over-day (Figure 3A), the WT mice clearly outperform the KI mice in days 2-7 and then WTs appear to rebound from their previous best performance on day 8. Eight training days is longer than the usual 6 days, and with the addition the four pre-trials with guidance to the platform by a rectangular channel, one could argue that this training schedule results in overtraining. Because WT outperform the KI mice on days 2-7 during training trials, additional experiments will be required to test performance in test trials on days 3 or 4 and day 6. Although decreased swim speed could in part explain the longer latencies for KI mice to reach platform, on day 8 it seems they had caught up arguing somewhat against the notion that the decrease in swim speed would fully explain the difference in training performance. It seems to me that the differences in latencies in Figure 3—figure supplement 1 actually are substantially larger than would be expected from the more modest reduction in swim speed shown in Figure 5A.

Third, that WT mice perform worse than the KI mice on day 1 is surprising. It would be important to analyze he trial-by-trial breakdown of this day, especially between trials 1/2 and 3/4, since 1/2 they were run several hours apart from 3/4, during which time memory could already begin to affect performance on subsequent trials.

Fourth, the quantification of discrimination of target vs. other quadrants in Figure 5C is unusual (one-way t-test against 25% chance line). A better depiction would be the usual analysis which shows time spent in each quadrant (not lumping all three non-target quadrants together) and perhaps also the distance accumulated in each quadrant due to the obvious difference in swim speed which will have consequences for dwell time in any region of the pool.

Finally, please clarify that heterozygous breeders were used to obtain litter matched WT and KI mice because even minor differences in the genetic background could affect performance. Also clarification is needed about the mice excluded for "procedural learning deficits" after they were run through training and the probe.

[Editors' note: further revisions were suggested prior to acceptance, as described below.]

Thank you for resubmitting your article "Long-term potentiation and spatial memory are independent of the C-tail of the GluA1 AMPA receptor subunit" for consideration by *eLife*. Your revised article has been considered by Linda Overstreet-Wadiche (Reviewing Editor) and Richard Aldrich as the Senior Editor.

The Reviewing Editor has drafted this decision to help you prepare a revised submission.

Summary:

This is an important study investigating a central question in synaptic neuroscience about mechanisms of AMPAR trafficking in LTP. Using three genetic approaches the authors convincingly show that the cytoplasmic tail of the GluA1 AMPAR subunit, which has been implicated in the past, is not essential for LTP. The reviewers agreed that the electrophysiological experiments are sophisticated, carefully executed, and convincing. However, the behavioral tests are not as thorough and extensive as needed to support the claim that spatial memory is independent of the C-terminal domain.

1) Essential Revisions Point 4. Please more specifically address this concern within the manuscript, since the topic of a modulatory role of GluA1 CTD on synaptic plasticity does not seem beyond the scope of the work. This could involve a more explicit explanation of possible scenarios that the authors consider to fall under the "receptor centric" model (as also mentioned in Pt 2) that might be helpful in differentiating the contrasting models for a general audience.

2) Essential Revisions Point 5. The authors have clarified a number of concerns related to MWM data, but there remains a confound related to swim speed. Due to Covid-19 issues the authors are unable to perform additional experiments to strengthen the overall conclusion about spatial memory. To better reflect the relative strength of the conclusions, agreed upon by all reviewers, the authors should remove reference to spatial memory (and MWM) in the Title and Impact Statement, and qualify "spatial memory assayed by the Morris Water Maze" (or similar) in the Abstract. As stated above, we are asking that the manuscript be revised to limit claims to those supported by data in hand, noting that potential confound to the current behavioral analysis requires additional supporting data from other behavioral tests that might be obtained in the future.

Title:

Please remove reference to spatial memory (and MWM) in the Title and Impact Statement

---

## [Author Response]

Essential revisions:1) Although the manuscript is clearly written, the authors need to provide more background to make this work accessible to a wider audience. In its current format one needs to be well-versed in the field and familiar with the Zhou et al. study to appreciate the new data. Please add explanation of “receptor slots”, more background on receptor-centric vs PSD-centric model (perhaps cite their recent review), the Malinow model, the role of receptor availability and the passive role for AMPARs in synaptic plasticity.

We appreciate the reviewer’s suggestion and have expanded the Introduction to frame our study in a broader perspective. This includes providing more detail on the previous studies that form the basis for the current study.

2) While the authors state that a widely held viewpoint is that covalent modifications of the GluA1 C-terminus results in GluA1 capture and anchoring at the postsynaptic site, this isn't necessarily the most prevalent model. Another view (not contradictory to the authors viewpoint that AMPARs in a reserve pool is critical for LTP) is that C-terminal modifications augment trafficking of GluA1-containing AMPAR to the postsynaptic site where via other mechanisms AMPAR become trapped to augment their postsynaptic functional availability. Framing the debate strictly in terms of receptor versus PSD centric models doesn't account for all views.

We appreciate the insightful comment from the reviewer. It is suggested that our framing of the debate is too simplistic and that, following the modification of GluA1, additional steps may be involved in trapping the receptors in the PSD. We would argue that the proposed scenario falls into the receptor centric model, since it is the modification of the receptor that triggers the trafficking to the PSD. Nevertheless, we agree with the reviewer in that several steps may be involved in trapping the receptors in the PSD. We have introduced a new paragraph in the Discussion where we explain our proposed model, based on recent findings, which establishes two essential requirements for AMPAR synaptic trafficking. First, the interactions between TARPs and PSD scaffolding proteins. Second, the presence of the extracellular AMPAR amino-terminal domain.

3) The text should be modified to clarify that the authors have not actually replicated the key experiments of Zhou et al., since they did not analyze the same mutations (replacing the GluA1 C-term with that of GLuA2 versus truncation) nor most of the same protocols (LTP of fEPSPs with inhibition intact, AMPAR conductance). Framing the current work primarily as addressing discrepancies with prior studies is not really satisfying when the discrepancies have not been explained.

We agree with the reviewer that, in order to satisfyingly address the discrepancies between our previous studies and the results reported by Zhou et al., we would ideally have had the chance to examine their KI mice, in which the CTD of GluA1 and GluA2 are swapped. We have repeatedly requested these mice to the corresponding authors of the Zhou et al. study beginning in December 2017. Our request has not been honored to date. Therefore, in our initial attempt to replicate the findings in Zhou et al., we used what we considered to be the closest alternative to the ideal tool (their mice). Identically to their manipulation, we replaced the CTD of GluA1 with that of GluA2 (Figure 1) using two different molecular replacement strategies in hippocampal CA1 pyramidal neurons through in utero electroporation of i) Cre recombinase together with GluA1/A2CTD and edited GluA2 (R) in Gria1-3^f/f^ mice and ii) Gria1 gRNA together with GluA1/A2CTD in Cas9 KI mice.

However, acknowledging the superiority of the KI approach, where the expression of chimeric or truncated AMPAR subunits is under the control of the endogenous promoter and gene expression regulators, we decided to turn to the KI approach. We determined that deleting the CTD of GluA1 was a better tool to address the role of that domain, since this approach prevents possible artifactual gain of function of the swapped CTD. Finally, we address the induction protocols used in the two studies and argue that our protocol closely mimics one of theirs.

4) There is no question that simultaneously mutating S831 plus S845 to alanine residues in the C-terminus of GluA1 in KI mice affects LTP induced by a so-called theta burst protocol as well as a pairing protocol, the latter similar to the protocols used in the current work (Lee et al., 2003, as cited). Apparently, when the GluA1 C-terminus is largely intact except for these two phosphorylation sites, then phosphorylation of one of the two sites is important for LTP (and learning as also tested in Lee et al., 2003). A possible explanation is that truncating the GluA1 C-terminus creates a situation in which such phosphorylation might be less important. For instance, perhaps non-phosphorylated GluA1 is held back from surface insertion or lateral diffusion to the postsynaptic site and phosphorylation of either S831 or S845 releases GluA1 to traffic to the postsynaptic site for anchoring there by means independent of S831 or S845 phosphorylation. When the C-terminus is removed as in the current manuscript than this inhibition is also removed. In order to put their results in context of prior work, the authors should discuss these possibilities and that the results depend on the exact mutations that are analyzed, and further the A2C-tail could also be an important factor.

We agree with the reviewer, and hope to have made it clear in our paper, that modification in the GluA1 might well have some modulatory effect on different physiological processes, including LTP. For example, we did find a significant decrease in swim speed in HA-ΔCTD GluA1 KI, consistent with previous findings with GluA1 KO mice (Zamanillo et al., 1999). These findings combined suggest that the regulation of swim performance by GluA1 requires its CTD. However, the main goal of our present study is to identify fundamental requirements of LTP, and our results using both molecular replacement and KI strategies argue strongly that the GluA1 CTD is not an absolute requirement for LTP. We feel that the numerous possible scenarios that might explain the modulatory role of the GluA1 CTD in previous results, as well as the possible role of the GluA2 CTD, are beyond the scope of this paper.

5) There are several issues with the behavior data that raise concerns with the strong conclusion about spatial memory. The authors should remove these data and claims, or provide additional experiments to make them more convincing.First, it would be more convincing to assess additional tests of spatial memory beyond MWM (such as novel object location and Y maze novel arm entry), especially in light of the swim speed confound.

We agree with the reviewer that more spatial memory tests would enhance the conclusion that spatial memory does not require the GluA1 CTD. Unfortunately, following the UCSF IACUC directives, we currently maintain colonies, including that of HA-ΔCTD GluA1 KI mice, to a minimal size. Hence, we do not currently have a large enough colony of HA-ΔCTD GluA1 KI mice to carry out the additional experiments proposed by this reviewer. In the current scenario, it would take many months to generate the required mice. We hope that this point-by-point response satisfies the reviewer’s concerns, and we are convinced that the behavior data that we present in this study is solid and justifies the conclusions that we reach. Since we have not been able to carry out further tests, we would be open to compromise and refer specifically to the water maze test instead of using the more general “spatial memory” in the title. We do believe that the spatial memory assessments in HA-ΔCTD GluA1 KI mice included in this paper constitute a valuable piece of information and we are, therefore, reluctant to remove the data.

Second, there are clear differences in performance during the training sessions in the MWM that warrant careful consideration. While the KI mice demonstrate gross learning day-over-day (Figure 3A), the WT mice clearly outperform the KI mice in days 2-7 and then WTs appear to rebound from their previous best performance on day 8. Eight training days is longer than the usual 6 days, and with the addition the four pre-trials with guidance to the platform by a rectangular channel, one could argue that this training schedule results in overtraining. Because WT outperform the KI mice on days 2-7 during training trials, additional experiments will be required to test performance in test trials on days 3 or 4 and day 6. Although decreased swim speed could in part explain the longer latencies for KI mice to reach platform, on day 8 it seems they had caught up arguing somewhat against the notion that the decrease in swim speed would fully explain the difference in training performance. It seems to me that the differences in latencies in Figure 3—figure supplement 1 actually are substantially larger than would be expected from the more modest reduction in swim speed shown in Figure 5A.

The reviewer has a valid point. However, as differences in the performance of mice on any given day can be influenced by many variables, examining individual training days is less informative in our experience than examining the overall learning curves, with the slope of the entire learning curve providing a robust measure of task acquisition/learning. When we compare the distance measures for the groups using a linear mixed model analysis, the most rigorous and robust statistical approach in our view, we obtain the following p values.

Days 1-6, *p* = 0.1252Days 1-7, *p* = 0.1313Days 1-8, *p* = 0.0807Days 2-7, *p* = 0.0839

Thus, truncating the analysis to 6 days of training, as suggested by the reviewer, actually increases the p value even more. The reviewer’s statement that “the WT mice clearly outperform the KI mice in days 2-7” is also not supported by this analysis, although there is a non-significant trend in this direction. The conclusion to be drawn is that WT and KI mice display spatial learning across the training regardless of how the data is parsed.

Third, that WT mice perform worse than the KI mice on day 1 is surprising. It would be important to analyze he trial-by-trial breakdown of this day, especially between trials 1/2 and 3/4, since 1/2 they were run several hours apart from 3/4, during which time memory could already begin to affect performance on subsequent trials.

When analyzing the data as suggested by the reviewer (see Author response image 1), it is clear that the performance of WT and KI mice was very similar by session 2 (trials 3/4). We therefore consider it unlikely that the difference between these groups in session 1 (trials 1/2) was due to learning, as the task was still unknown to the mice at that time.

Fourth, the quantification of discrimination of target vs. other quadrants in Figure 5C is unusual (one-way t-test against 25% chance line). A better depiction would be the usual analysis which shows time spent in each quadrant (not lumping all three non-target quadrants together) and perhaps also the distance accumulated in each quadrant due to the obvious difference in swim speed which will have consequences for dwell time in any region of the pool.

We apologize for the confusion regarding the statistics used in Figure 5C. The blue line on the graph at 25% was added to illustrate chance performance but was not used for statistical analysis. We performed paired t-test analysis on this data comparing % time in the target quadrant to the average % time in the 3 non-target quadrants for each of the genotypes. As the reviewer mentions, displaying the % time spent in each quadrant is confounded by the slower swim speeds in the KI animals. Therefore, focusing on the path length in each quadrant gives a better approximation of quadrant preference in the present study. We show below the distances the mice swam in each quadrant during the probe trial (see Author response image 2).

A recent publication warned against comparing groups of mice based on statistical analyses carried out strictly within rather than across groups (Nygaard et al., 2019, *Autism Research*). To allow for a more direct comparison of WT and KI mice in the probe trial, we divided the path length in the target quadrant by the average path length in the non-target quadrants for each mouse, and then used a Mann-Whitney U test to compare the target preference ratios between groups. This analysis, shown in Author response image 2, confirms comparable target preferences between WT and KI mice (p = 0.348).

**Author response image 2. respfig2:** 

Finally, please clarify that heterozygous breeders were used to obtain litter matched WT and KI mice because even minor differences in the genetic background could affect performance. Also clarification is needed about the mice excluded for "procedural learning deficits" after they were run through training and the probe.

As stated in the Materials and methods section, both littermate and cage mate male mice were used for behavior experiments. The breeding strategy, designed in consultation with the Gladstone Behavior Core, which has extensive experience in these experiments, included both heterozygous x heterozygous breedings and KI and WT homozygous x homozygous breedings. We agree with the reviewer that it is preferable that all mice are littermates. Given a 25% chance of homozygous WT or KI pups from a heterozygous breeding (12.5% of male homozygous pups) and a litter size around 7 pups, which is typical for the C57Bl6 strain, we obtain less than one homozygous WT + KI male mice pair per heterozygous breeding litter. Hence, to ensure that we obtained a sufficient cohort of mice of both genotypes around the same age, we also used homozygous breedings. We now better clarify that both types of breedings were used to generate the mice included in the study in the Materials and methods section.

One KI mouse was excluded from the probe trial due to extreme floating behavior and two mice (one from each genotype) were excluded from both training and probe trial analyses because they showed no evidence for task acquisition at all. Exclusions were done blind to the genotype.

[Editors' note: further revisions were suggested prior to acceptance, as described below.]

Revisions for this paper:1) Essential Revisions Point 4. Please more specifically address this concern within the manuscript, since the topic of a modulatory role of GluA1 CTD on synaptic plasticity does not seem beyond the scope of the work. This could involve a more explicit explanation of possible scenarios that the authors consider to fall under the "receptor centric" model (as also mentioned in Pt 2) that might be helpful in differentiating the contrasting models for a general audience.

In the revised manuscript, we have included a more extensive discussion of plausible reasons why we have been unable to find a modulatory role for the GluA1 CTD in our study, in line with the suggestions from reviewers. We have also provided more detail to the explanation of the two contrasting models of LTP.

2) Essential Revisions Point 5. The authors have clarified a number of concerns related to MWM data, but there remains a confound related to swim speed. Due to Covid-19 issues the authors are unable to perform additional experiments to strengthen the overall conclusion about spatial memory. To better reflect the relative strength of the conclusions, agreed upon by all reviewers, the authors should remove reference to spatial memory (and MWM) in the Title and Impact Statement, and qualify "spatial memory assayed by the Morris Water Maze" (or similar) in the Abstract. As stated above, we are asking that the manuscript be revised to limit claims to those supported by data in hand, noting that potential confound to the current behavioral analysis requires additional supporting data from other behavioral tests that might be obtained in the future.

References to spatial memory and MWM have been removed from the Title and Impact Statement. An explicit acknowledgement of the possible confounding effect of the reduced swim speed in the KI, and that future research will be needed to fully understand the role of the GluA1 CTD in spatial memory, has been included.

Title:Please remove reference to spatial memory (and MWM) in the Title and Impact Statement

We have removed the reference to spatial memory and MWM from the Title and Impact Statement.